# New Imaging Markers of Clinical Outcome in Asymptomatic Patients with Severe Aortic Regurgitation

**DOI:** 10.3390/jcm8101654

**Published:** 2019-10-11

**Authors:** Radka Kočková, Hana Línková, Zuzana Hlubocká, Alena Pravečková, Andrea Polednová, Lucie Súkupová, Martin Bláha, Jiří Malý, Eva Honsová, David Sedmera, Martin Pěnička

**Affiliations:** 1Department of Cardiology, Institute for Clinical and Experimental Medicine, Prague 14021, Czech Republic; 2Faculty of Medicine in Hradec Králové, Charles University, Šimkova 870, Hradec Králové 500 03, Czech Republic; 3Department of Cardiology, Royal Vinohrady University Hospital, Prague 10034, Czech Republic; 4Department of Cardiology, General University Hospital, Prague 12808, Czech Republic; 5Department of Cardiothoracic surgery, Institute for Clinical and Experimental Medicine, Prague 14021, Czech Republic; 6Institute for Clinical and Experimental Medicine, Clinical and Transplant Pathology Centre, Prague 14021, Czech Republic; 7First Faculty of Medicine, Institute of Anatomy, Charles University in Prague, Prague 12800, Czech Republic; 8Cardiovascular Center Aalst, OLV Clinic, 9300, Belgium

**Keywords:** aortic regurgitation, echocardiography, magnetic resonance imaging, vena contracta area, longitudinal strain, T1 mapping

## Abstract

*Background:* Determining the value of new imaging markers to predict aortic valve (AV) surgery in asymptomatic patients with severe aortic regurgitation (AR) in a prospective, observational, multicenter study. *Methods:* Consecutive patients with chronic severe AR were enrolled between 2015–2018. Baseline examination included echocardiography (ECHO) with 2- and 3-dimensional (2D and 3D) vena contracta area (VCA), and magnetic resonance imaging (MRI) with regurgitant volume (RV) and fraction (RF) analyzed in CoreLab. *Results:* The mean follow-up was 587 days (interquartile range (IQR) 296–901) in a total of 104 patients. Twenty patients underwent AV surgery. Baseline clinical and laboratory data did not differ between surgically and medically treated patients. Surgically treated patients had larger left ventricular (LV) dimension, end-diastolic volume (all *p* < 0.05), and the LV ejection fraction was similar. The surgical group showed higher prevalence of severe AR (70% vs. 40%, *p* = 0.02). Out of all imaging markers 3D VCA, MRI-derived RV and RF were identified as the strongest independent predictors of AV surgery (all *p* < 0.001). *Conclusions:* Parameters related to LV morphology and function showed moderate accuracy to identify patients in need of early AV surgery at the early stage of the disease. 3D ECHO-derived VCA and MRI-derived RV and RF showed high accuracy and excellent sensitivity to identify patients in need of early surgery.

## 1. Introduction

Chronic aortic regurgitation (AR) is the third most common valvular heart disease in Western countries, with a prevalence of 0.1% to 2.0%. Degenerative etiology on tricuspid or bicuspid aortic valves and annuloaortic ectasia are the most common causes of chronic AR. Rheumatic fever remains a frequent cause of chronic AR in developing countries [1,2,3,4]. In chronic AR, progressive left ventricular (LV) dilatation compensates for the increase of LV end-diastolic pressure. Left ventricular dilatation enables the preservation of cardiac output despite the large regurgitant volume of blood returning to the LV in diastole. Increased afterload in chronic AR is compensated by eccentric LV hypertrophy. This complex volume and pressure compensatory mechanisms explain the long asymptomatic course of the disease.

A combination of several clinical and imaging characteristics makes the timing of aortic valve (AV) intervention challenging. Patients with severe AR are often middle-aged males with a long asymptomatic course [1,2,5]. Furthermore, current indications for AV intervention (i.e., two-dimensional (2D) echocardiography (ECHO)-derived assessment of AR severity and left ventricular (LV) remodeling), appear to be rather specific but less sensitive [1,2,6,7,8,9,10]. This may lead to late AV intervention resulting in irreversible myocardial damage and impaired outcome [1,2,3,11,12]. Thus, a more a sensitive and accurate imaging marker to trigger early AV intervention may be of crucial importance.

Several promising imaging approaches to assess AR or its impact on LV structure and function have emerged recently [13,14,15,16,17,18,19,20]. For instance, three-dimensional (3D) ECHO- or magnetic resonance (MRI)-derived evaluation of AR severity may increase the accuracy of AR classification [16,19,20]. Assessment of LV myocardial fibrosis, strain or work may be more sensitive than LV diameters or ejection fraction (LVEF) to detect irreversible myocardial damage at an early stage [13,14,15,17,18,21]. Therefore, in a multicenter study, we thought to determine the value of new imaging markers to predict AV surgery in asymptomatic patients with severe AR and preserved LVEF. Imaging markers from all participating centers were analyzed centrally by a CoreLab.

## 2. Experimental Section

### 2.1. Design

A prospective, observational and multicenter study was conducted in three tertiary cardiology centers. All imaging markers were evaluated centrally in a CoreLab (Institute of Clinical and Experimental Medicine, Prague), which holds the European Association of Cardiovascular Imaging (EACVI) Laboratory accreditation and individual certification for both ECHO and MRI.

### 2.2. Patients 

The study population consisted of all consecutive patients (age 44.4 ± 13.2 years, 85.4% males) with chronic severe AR and no indication for AV intervention who were referred to participating heart valve centers for AR assessment between March 2015 and September 2018. To be eligible for the study patients had to fulfil the following inclusion criteria: (1) severe AR defined by using the integrative 2D ECHO approach [10]; (2) absence of symptoms validated using bicycle ergometry; (3) preserved LVEF (>50%); (4) non-dilated LV end-diastolic diameter (≤70 mm) and LV end-systolic diameter index (≤25 mm/m^2^); and (5) sinus rhythm. Patients with guideline indications for AV intervention, acute AR, aortic dissection, endocarditis, irregular heart rate, associated with more than mild valvular disease, complex congenital heart disease, intracardiac shunt, creatinine clearance <30 mL/min, pregnancy, or contra indication for MRI were excluded [1,2]. The study protocol and informed consent was approved by the ethics committees of all participating institutions. All patients had to sign informed consent prior to the enrollment. The study was registered in ClinicalTrials.gov under a unique identifier NCT02910349.

### 2.3. Protocol

An initial assessment was performed by specialized heart valve cardiologist in all participating centers. It included a history, clinical examination, Electrocardiography (ECG), bicycle ergometry, blood sampling, and comprehensive 2D and 3D ECHO. A cardiac MRI was performed in the center where the CoreLab was based for all participants. Patients from this particular center underwent MRI on the day of enrollment while patients from other two centers underwent an MRI within 2 weeks after the enrollment. Analysis of all ECHO- and MRI-derived markers were centralized in the CoreLab. 

### 2.4. Follow-Up and Study Endpoints

After enrollment, patients were followed in participating heart valve centers every 6 months till 30 September 2018. The decision-making on conservative versus surgical treatment was left at the discretion of a particular heart valve team. In patients undergoing AV surgery, a perioperative biopsy was performed at the level of basal interventricular septum to assess the extent of myocardial fibrosis as previously described in our pilot study [14]. The follow-up data on AV interventions, mortality, and cardiac hospitalizations were obtained in all patients (100%) using population registry, medical files, and contact with referring physicians or family. Baseline clinical and imaging characteristics were analyzed to identify independent predictors of AV surgery. A prespecified study endpoint was cumulative of the indication for aortic valve intervention, ventricular arrhythmia occurrence (non-sustained or sustained ventricular tachycardia, ventricular ectopic beats >10%), hospitalization for heart failure, Brain natriuretic peptide (BNP) elevation >150 ng/L or cardiovascular death. 

### 2.5. Doppler ECHO 

A comprehensive 2D and 3D transthoracic ECHO was performed using a Vivid 7 and Vivid 9 (GE HealthCare, Horten, Norway) ultrasound system equipped with 4-dimensional active matrix 4-D volume phased array probe. Several 3D ECHO loops in each view were recorded using ECG-gated full-volume acquisition over four (LV function) to six (color Doppler) cardiac cycles during end-expiratory apnea. Images were optimized by adjusting the depth, sector size, gain, number of frames per second (FPS), the number of heart beats, and breath hold. All acquired images were digitally stored, anonymized, and analyzed using the commercially available software EchoPac BT 202, GE HealthCare. An average of at least 3 beats was taken for each measurement. Blood pressure and heart rate was recorded during each examination.

LV internal diameters were derived from an LV internal cavity using M-mode whenever possible. The biplane Simpson method was used to assess 2D LV volumes and Ejection fraction (EF) [21]. A semi-automatic contouring method with manual correction was used to measure 3D LV volumes and EF [21]. Global longitudinal strain (GLS) was assessed using a semi-automatic speckle tracking method with manual adjustment with frame rate of >60 FPS for 2D GLS (Figure 1B) and LV twist, and >25 FPS for 3D GLS [21]. Myocardial work was derived from 2D GLS, brachial blood pressure, and the timing of valvular events as described previously [18,22]. The severity of AR was assessed using the recommended approach integrating valve morphology, vena contracta width, the size of regurgitant jet in LV cavity and its width in LVOT, the jet pressure half time, the velocity of the diastolic flow reversal in the descending aorta, and the size of the proximal isovelocity surface area (PISA) [7,10]. Given the high prevalence of bicuspid valves in the study population and eccentric jets, the calculation of regurgitant volume (RV) using the PISA method was not consistently feasible. Therefore, the RV and regurgitant fraction (RF) of AR was assessed using the Doppler volumetric method, which uses the differences between the mitral and aortic stroke volumes (SV) to calculate RV and RF of AR [7,10]. Moderate-to-severe AR was defined by the presence of 2–3 specific criteria and RV of 45–59 mL or RF of 40%–49% [7,10]. Severe AR was defined by the presence of ≥4 specific criteria or by the presence of 2–3 specific criteria and RV ≥60 mL or RF ≥50% [10]. Moreover, 3D ECHO derived vena contracta area (VCA) was assessed in zoomed parasternal long-axis view (Figure 1A). In brief, the narrowest sector possible and multibeat acquisition was used to maximize the frame rate. To identify VCA, the 3D dataset was rotated to bisect the regurgitant color jet at the level of the leaflet coaptation zone perpendicularly to its long axis in 2 orthogonal planes. The image was cropped along the jet direction to visualize the cross-sectional area at the level of the vena contracta. Low velocity peripheral signals of the color spectrum were rejected. The VCA was defined as the high velocity core of the color spectrum [20].

### 2.6. Cardiac MRI 

An examination was performed using a 1.5T scanner (Magnetom Avanto fit, Siemens, Munich, Germany). The protocol consisted of pilots, T2 weighted dark blood, cine with 2-, 4-, 3-chambers, and short axis (SA) images covering the entire end-diastolic ventricular length, through-plane phase-contrast velocity mapping at the level of the aortic root, contrast late gadolinium enhancement (LGE), and modified Look–Locker Inversion recovery sequence (MOLLI) [14,23]. Analysis was performed using a commercially available software (Segment CMR, Medviso AB 2018, Lund, Sweden). Blood pressure and heart rate was recorded during each examination.

LV volumes and LVEF were calculated using the steady state free precession cine imaging in short-axis stack (slice thickness 8 mm, slice spacing 0) with correction for the valve position in long-axis planes. LV radial, circumferential and longitudinal strain were assessed in the short axis and apical views. Native T1 relaxation time and extracellular volume fraction (ECV) were evaluated using the MOLLI sequence as previously described [14]. The T1 relaxation time measurement was performed in basal-to-mid short-axis slice 15 min pre and 15 min post contrast administration (Figure 1F). Parameters of the MOLLI sequence were as follows: field of view (FoV) 360 × 301 mm, matrix 118 × 256, slice thickness 8 mm, voxel size 1.4 mm × 1.4 mm × 8 mm, echo time 1.1 ms, repetition time 359 ms, flip angle 35°, bandwidth 1.085 Hz/Pixel.

Aortic forward and regurgitation flow were obtained by using the through-plane phase-contrast velocity mapping during breath-hold over 10–20 ms with retrospective ECG gating (Figure 1D). Parameters were as follows: temporal resolution 25–55 ms; echo time 2.7 ms; repetition time 46.8 ms, FoV 300 × 200 mm; matrix size 192 × 132; velocity window 1.5 to 4.0 m/s. Several image slices were prescribed at the level of the aortic root in end-diastole starting from 0.5 cm above AV annulus to 0.5 cm above the sinotubular junction (slice thickness 6 mm, spacing 0). Care was taken to align the slices perpendicularly to the direction of blood flow in two orthogonal imaging planes (Figure 1C). The lowest velocity encoding without flow aliasing was chosen for the analysis. Background velocity offset errors were corrected by using flow stationary phantom and post-processing correction. It has been shown previously that the ascending aorta slice location with the most accurate measurement of RV is between annulus and coronary ostia [24]. To reduce underestimation of RV and RF of AR, the closest slice to the AV without interference of turbulent flow was selected for the analysis [16,19]. Flow measurements from 3 acquisitions were averaged. Aortic forward volume (SV) and RV were derived by integration of the flow curve over 1 cardiac cycle. RF was calculated as RV/(SV * 100%) (Figure 1E).

### 2.7. New Imaging Markers 

Apart from conventional clinical and imaging parameters, the value of the following new imaging markers to predict AV surgery was tested: 2D GLS, 2D myocardial work, 3D GLS, 3D VCA, MRI-derived native T1 relaxation time and ECV, MRI-derived GLS and circumferential strain, and MRI-derived RV and RF.

### 2.8. Statistical Analysis 

Data are expressed as mean ± SD for continuous variables and as counts or percentages for categorical variables. Unpaired Student *t*-test, Pearson χ^2^ or Fisher exact tests were used as appropriate. Receiver-operating characteristic curve analysis was used to identify imaging markers to predict future AV surgery. The optimal cutoff value for sensitivity and specificity was calculated according to the Youden’s index and according to the clinical relevance. Several Cox proportional hazard models were used to identify independent predictors of aortic valve surgery (AVR). The selection of variables for the models was based on clinical relevance. Care was taken to avoid overfitting and to avoid combining mutually dependent variables in one analysis. Results were reported as hazard ratios (HRs) with the 95% of confidence interval (95% CI) of probability values. The Kaplan–Meier method and log-rank test was used for temporal analysis of differences in AV surgery between groups. For all tests, values of *p* < 0.05 were considered significant. Statistical analysis was performed using the SPSS version 20 (SPSS Inc, Chicago, IL, USA) and the GraphPad Prism version 6.0 (GraphPad Software, San Diego, CA, USA).

## 3. Results

### 3.1. Baseline Clinical and Imaging Characteristics

Table 1 and Table 2 show baseline clinical and imaging characteristics, respectively. A two-dimensional ECHO study with satisfactory image quality was successfully completed in all patients. Feasibility of 3D ECHO was 99% for LV volumes, 89% for 3D GLS, and 100% for VCA. Three patients (3%) failed to complete the cardiac MRI because of claustrophobia (*n* = 2) or severe spine deformity (*n* = 1). Feasibility of MRI-derived T1 mapping and LV strain was 96% and 95%, respectively. Blood pressure and heart rate was similar during ECHO and MRI examination. The majority of patients were middle-aged males (86%) with bicuspid AV (76%). The most prevalent risk factor for coronary artery disease was hypertension (47%) with corresponding medication. Per study inclusion criteria, all patients were asymptomatic, in sinus rhythm, with normal LV dimensions, and LVEF. A total of 56 (54%) individuals had moderate-to-severe AR while the remaining 48 (46%) showed severe AR. During median follow-up of 587 days (interquartile range (IQR) 296–901 days), no patient died. A total of 20 (19%) individuals underwent AV surgery (surgical group) while the remaining patients were treated conservatively (conservative group). Median time to AV surgery was 236 days (IQR 125–460 days). Clinical characteristics, BNP, and creatinine values were similar between groups (Table 1).

### 3.2. Assessment of LV Morphology and Function

At 2D ECHO, patients who underwent AV surgery had significantly larger LV dimensions, end-diastolic volume (LVEDV) (all *p* < 05) and tended to have larger end-systolic volume (LVESV) than patients treated conservatively (Table 2). MRI-derived LV volumes showed a similar trend with significantly larger volumes in the surgical versus the conservative group (all *p* < 0.01). In contrast, LVEF derived by using whatever method was similar. Imaging markers of subtle myocardial damage (i.e., the MRI-derived T1 relaxation time or ECV, 2D-, 3D-, or MRI-derived LV strain), did not show statistically significant differences between groups, although 2D GLS tended to be lower in the surgical versus the conservative group (*p* = 0.07). Average myocardial fibrosis on perioperative myocardial biopsy (*n* = 14) was 15 ± 20%. The degree of myocardial fibrosis correlated significantly with MRI-derived LV mass (r = 0.66), native T1 relaxation time (r = 0.56), ECV (r = 0.31), and 2D ECHO-derived GLS (0.46).

### 3.3. Assessment of AR Severity

Using the integrative approach, the surgical group showed a significantly higher prevalence of severe AR than the conservative group (70% vs. 40%, *p* = 0.02). Accordingly, we observed significantly larger velocity of the diastolic flow reversal in the descending aorta, 2D ECHO RV and RF of AR, and 3D ECHO VCA in patients treated surgically versus conservatively (all *p* < 0.05) (Table 2). In contrast, 2D vena contracta width was similar between groups. MRI-derived RV and RF were significantly larger in patients undergoing surgery than in patients treated conservatively (both *p* < 0.01).

### 3.4. Prediction of AV Surgery

Table 3 and Figure 2 show accuracy of selected imaging markers to identify patients who underwent AV surgery. The integrative 2D ECHO approach had high negative predictive value (89%), but low positive predictive value (29%) to identify future AV surgery. All the ECHO- and MRI-derived indices of LV remodeling and function had an area under the curve (AUC) <0.7. Out of the ECHO-derived parameters, end-systolic diameter (LVESD), with an optimal cutoff value of >37 mm, its index (LVESDi), with an optimal cutoff value of >18 mm/m^2^, had the largest AUC. Higher cutoff values of LVESD (>45 mm) or LVESDi (>22 mm/m^2^), approaching the guideline recommendations, were highly specific (>90%) but lacked the sensitivity (<20%). The MRI-derived volumes and their indexed values were also rather specific than sensitive. Out of the ECHO-derived indices of AR severity, the largest AUC was observed for velocity of diastolic flow reversal in aorta descendens and 3D VCA. The optimal cutoff value of 3D VCA ≥30 mm^2^ had a sensitivity of 80% and a specificity of 63% to identify future AV surgery (Figure 3A). A total of 31 patients treated conservatively had VCA ≥30 mm^2^ (false positive). Combining VCA with LVESD or LVESDi increased the specificity up to 97% depending on the cutoff (Table 3). Out of all tested imaging markers, the MRI-derived RV, with a cutoff value ≥45 mL (Figure 3B), and the MRI-derived RF, with a cutoff value ≥34% (Figure 3C), showed the largest AUC (>0.75) with very high sensitivity (≥90%). A total of 33 out of 36 patients in the conservative group had RV ≥45 mL and RF ≥34%, respectively (false positive). Combining RV and RF with LV end-diastolic or end-systolic volume index increased the specificity up to 78% and 89%, respectively (Table 3). In Cox regression analysis, 3D VCA, MRI-derived RV and RF were identified as strongest independent predictors of AV surgery (Table 4). In contrast, LV strain, T1 time or ECV were not independent predictors.

## 4. Discussion

The present study included asymptomatic patients with normal LVEF and non-dilated LV. Compared with the previous reports in asymptomatic AR patients, individuals included in the present study were younger, had less dilated LV, lower BNP, higher magnitude of GLS, and experienced less endpoints during follow-up [6,8,13,15,19,25]. This suggests the very early stage of AR disease. The findings of the present study can be summarized as follows: (1) AR severity seemed to be the major determinant of early disease progression while indices of LV morphology and function showed lower predictive accuracy; (2) new imaging markers of AR severity (i.e., 3D VCA, MRI-derived RV and RF), showed higher sensitivity than those derived using 2D Doppler ECHO; (3) integrating a sensitive with a specific parameter, for instance ECHO-derived VCA with LVESDi, or MRI-derived RV or RF with LVEDVi or LVESVi showed higher discriminative power than 2D ECHO integrative approach to identify patients undergoing early AV surgery.

### 4.1. LV Morphology and Function 

Chronic AR leads to LV volume and pressure overload with subsequent hypertrophy, dilatation, systolic dysfunction, and heart failure. LV dimensions (LVESD >50 mm or LVESDi >25 mm/m^2^), and LVEF (LVEF <50%) are currently used as indications for AV intervention [1]. Several recent studies demonstrated low sensitivity of these cutoffs by showing improved outcome in patients who had been operated on before the onset of these triggers [6,8]. In the present study, the optimal cutoff of LVEDSi (>18 mm/m^2^), with acceptable sensitivity (80%), was lower than previously proposed [1,6,8]. Using higher cutoff of 20 or 22 mm/m^2^ increased specificity (75%–93%) at the expense of unacceptably low sensitivity (15%–30%). MRI-derived volumes were rather specific but had lower sensitivity. It is of note, that a considerable proportion of patients (38%) showed increased LV volumes at MRI despite normal 2D ECHO dimensions. Nevertheless, the predictive accuracy of LV dimensions or volumes, derived by either technique, were moderate with an area under the curve <0.7 in all cases. Several new markers describing subtle myocardial damage or dysfunction have emerged recently [13,14,15,16,17,18,19,20]. MRI-derived native T1 mapping and ECV are accurate and validated markers of diffuse myocardial fibrosis [14,23,26]. ECHO-derived GLS has been introduced as a sensitive marker of early systolic dysfunction and potentially of clinical outcome in different valvular diseases [1,21]. Several studies reported independent association between speckle-tracking-derived GLS and the need for AV surgery [13,15,17]. In the present study, only 2D GLS tended to be lower in the surgical versus the conservative group while 3D GLS or MRI-derived strains, T1 relaxation time, and ECV were similar. The explanation of different findings can be that previous studies included more advanced disease as documented by a higher prevalence of endpoints, older age, more dilated LV or lower magnitude of GLS compared with our data [13,15]. Of interest, in the surgical group, we observed increased myocardial fibrosis (median 15%) at perioperative biopsy. These values are clearly elevated as a normal range between 1%–4.5% [27,28]. Both T1 relaxation time, ECV, and GLS showed significant correlation with the extent of fibrosis in histological samples. Moreover, T1 relaxation time was significantly longer (1022 ± 30 ms vs. 980 ± 22 ms, *p* < 0.01) and 2D GLS significantly lower (−18 ± 2% vs. −22.5 ± 2%, *p* < 0.01) compared with 30 healthy controls. Yet, these parameters failed to identify patients with early disease progression. This suggests that at the early stage of AR disease, the parameters reflecting subtle myocardial damage may not be accurate enough to predict early disease progression. 

### 4.2. Assessment of AR Severity

The majority of recommended indices to assess AR are semiquantitative, lack the sensitivity or their accuracy is hampered by jet eccentricity [7,9,10]. Accordingly, in the present study with high prevalence bicuspid AV and eccentric jets, the consistent measurement of PISA-derived effective regurgitant orifice (ERO) and RV was not possible. It might have been for the same reasons that 2D vena contracta width did not show significant differences between groups. In contrast, 3D data can be rotated perpendicular to the jet direction in several planes to avoid the limitation introduced by jet eccentricity. The vena contracta area is a 3D-derived area of the vena contracta without any geometric assumption. VCA has been shown to be highly accurate, reproducible, and superior to the PISA method in different native valve regurgitations [20,29,30,31]. In the present study, 3D VCA had the highest accuracy out of all ECHO markers of AR severity to identify patients in need for early AV surgery. A combination of sensitive VCA with specific LVEDSi showed the optimal discriminative power. MRI-derived assessment of blood flow at the level of the aortic root is a highly reproducible and quantitative technique, which allows for direct assessment of RV of AR [16,19]. In the present study, MRI-derived RV and RF showed the largest accuracy out of all the imaging parameters to predict AV surgery. Our cutoff values of RV (≥45 mL) and RF (≥34%) were similar to values (RV >42 mL, RF >33%) published previously in more advanced AR disease [19]. In our study, both RV and RF were highly sensitive but less specific. In contrast, Myerson observed balanced high sensitivity (92%–85%) and specificity (85%–92%) for both indices [19]. This difference in specificity may be related to the very early stage of AR disease in our study while Myerson included older patients with more a dilated LV [19]. In the present study, combining sensitive RV or RF with specific LV volumes or their indices increased the specificity to identify future AV surgery. Of note, 2D ECHO integrative approach showed lower predictive accuracy. This suggests that, in asymptomatic patients with severe AR, both 3D ECHO-derived VCA and MRI-derived RV and RF may be clinically useful to increase sensitivity and accuracy of the recommended approach.

## 5. Conclusions

The present study assessed the clinical value of new imaging markers in asymptomatic patients with chronic severe AR at the early stage of the disease. Parameters related to LV morphology and function showed moderate accuracy to identify patients in need for early AV surgery. This suggests their limited accuracy at the early stage of AR disease while they may become useful later in the disease course with ongoing LV remodeling. In contrast, 3D ECHO-derived VCA and MRI-derived RV and RF showed the highest accuracy and excellent sensitivity to identify patients in need for early AV surgery. This suggests their clinical potential since the recommended integrative approach is rather specific than sensitive.

## Figures and Tables

**Figure 1 jcm-08-01654-f001:**
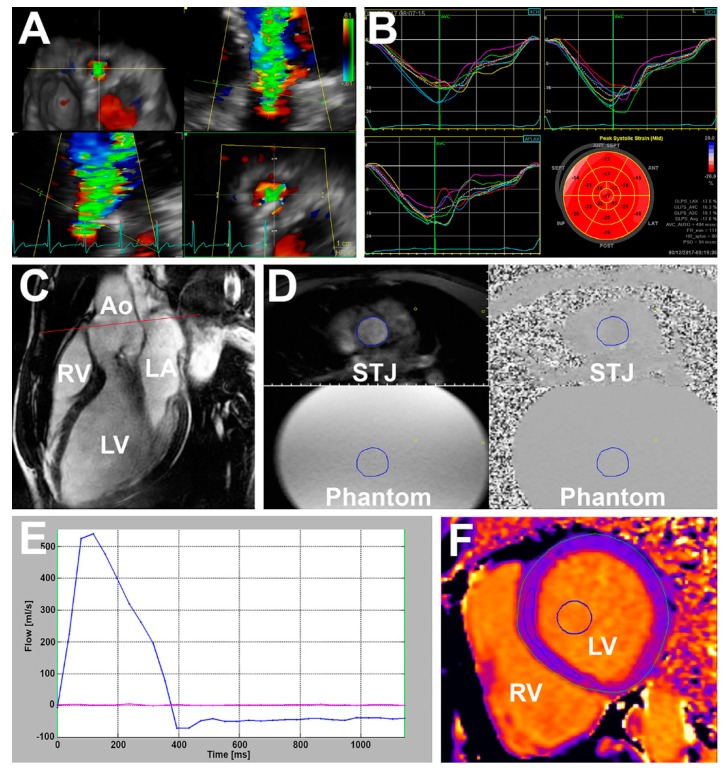
Imaging markers. (**A**) Echocardiography derived three-dimensional vena contracta area; (**B**) echocardiography two-dimensional global longitudinal strain; (**C**) magnetic resonance—the left ventricular outflow tract (cine), red line—through-plane flow sequence slice position displayed on, Ao—aorta, LA—left atrium, LV—left ventricle, RV—right ventricle; (**D**) through-plane flow sequence at sinotubular junction level (STJ) of the aorta (displayed on (**C**)), the blue circle is a manually drawn region of interest where the blood flow and regurgitant volume and fraction are calculated. The exact copy of the region interest is in all four images, phantom—stationary phantom used for flow measurement correction; (**E**) flow-time curve based on (**D**)—blue line shows blood flow at STJ and red line show flow in stationary phantom; (**F**) native T1 mapping from modified Look–Locker Inversion recovery sequence (MOLLI) sequence, blue circle—a semi-automatically drawn region of interest within the blood pool, blue ellipsoid—a manually drawn region of interest within the myocardium at the level of the interventricular septum utilized for myocardial fibrosis calculation.

**Figure 2 jcm-08-01654-f002:**
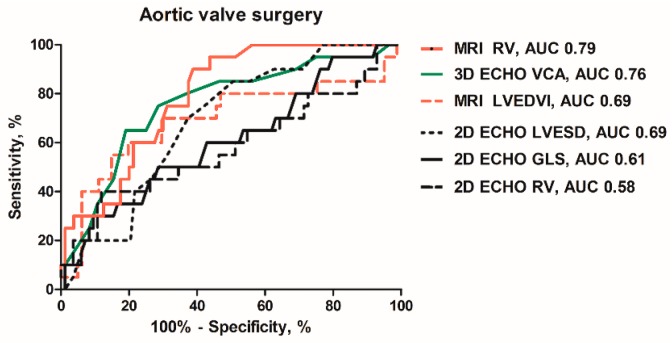
Receiver-operating characteristics curves of the MRI-derived: regurgitant volume (RV) and left ventricular end-diastolic volume index (LVEDVI); the 3D ECHO-derived: vena contracta area (VCA); 2D ECHO-derived: left ventricular end-systolic diameter (LVESD); RV and global longitudinal strain (GLS) to predict AV surgery.

**Figure 3 jcm-08-01654-f003:**
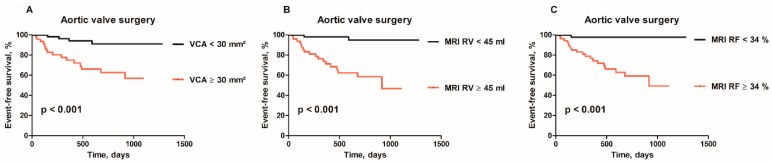
(**A**) Kaplan–Meier curves for aortic valve surgery (AVR) in patients with 3D ECHO-derived VCA ≥30 mm^2^ vs. <30 mm^2^, (**B**) MRI-derived RV ≥45 mL vs. <45 mL; and (**C**) MRI-derived RF ≥34% vs. <34%.

**Table 1 jcm-08-01654-t001:** Baseline clinical characteristics.

	Total (*n* = 104)	Conservative (*n* = 84)	Surgical (*n* = 20)	*p*-Value
Age, years	44 ± 13	44 ± 13	45 ± 14	0.922
Male gender, N (%)	89 (86)	72 (86)	17 (85)	0.292
Hypertension, N (%)	50 (48)	40 (48)	10 (50)	0.801
Diabetes mellitus, N (%)	6 (6)	5 (6)	1 (5)	1.000
Hyperlipidemia, N (%)	29 (28)	24 (29)	5 (25)	0.773
Smoker, N (%)	14 (13)	11 (13)	3 (15)	1.000
Coronary artery disease, N (%)	4 (4)	3 (4)	1 (5)	0.542
Previous cardiac surgery, N (%)	4 (4)	3 (4)	1 (5)	1.000
Stroke, N (%)	1 (1)	0 (0)	1 (5)	0.175
Aspirin, N (%)	10 (10)	7 (8)	3 (15)	0.686
Oral anticoagulants, N (%)	5 (5)	3 (4)	2 (10)	0.542
ACEI/ARBs, N (%)	54 (52)	44 (52)	10 (50)	0.607
Beta-blockers, N (%)	25 (24)	21 (25)	4 (20)	0.231
Calcium channel blockers, N (%)	20 (19)	16 (19)	4 (20)	0.759
Diuretics, N (%)	15 (14)	9 (11)	6 (30)	0.261
Statins, N (%)	22 (21)	18 (21)	4 (20)	1.000
NYHA Class I, N (%)	104 (100)	84 (100)	20 (100)	1.000
Height, cm	180 ± 9	180 ± 9	181 ± 8	0.752
Weight, kg	85 ± 14	84 ± 14	86 ± 14	0.744
Systolic blood pressure, mmHg	136 ± 16	135 ± 16	139 ± 18	0.334
Diastolic blood pressure, mmHg	70 ± 12	71 ± 12	68 ± 12	0.292
Heart rate, beats per min	64 ± 10	63 ± 10	64 ± 13	0.965
Sinus rhythm, N (%)	104 (100)	84 (100)	20 (100)	1.000
B-natriuretic peptide, ng/L	27 (42)	24 (36)	34 (117)	0.054
Creatinine Clearance mL/min	118 ± 31	120 ± 32	107 ± 28	0.110
*Aortic valve morphology*				0.551
Trileaflet, N (%)	14 (13.6)	11 (12.9)	3 (16.7)	
Bicuspid, N (%)	79 (76.7)	65 (76)	14 (70)	
Unicuspid/quadricuspid, N (%)	4 (4)	4 (5)	0 (0)	
Unknown, N (%)	6 (6)	4 (5)	2 (10)	

Values are means ± standard deviations, median (interquartile range) or numbers (percentage). ACEI/ARB, angiotensin converting enzyme inhibitor/angiotensin receptor blocker; NYHA, New York Heart Association.

**Table 2 jcm-08-01654-t002:** Baseline imaging characteristics.

	Total (*n* = 104)	Medical (*n* = 84)	Surgical (*n* = 20)	*p*-Value
*LV assessment*				
2D ECHO end-diastolic diameter, mm	58 ± 6	58 ± 6	61 ± 4	0.031
2D ECHO end-systolic diameter, mm	37 ± 5	37 ± 5	40 ± 4	0.006
2D ECHO end-systolic diameter index, mm/m^2^	18 ± 3	18 ± 3	20 ± 3	0.019
2D ECHO end-diastolic volume, mL	158 ± 68.0	156 ± 58	194 ± 60	0.008
2D ECHO end-diastolic volume index, mL/m^2^	77 ± 31	76 ± 26	89 ± 32	0.019
2D ECHO end-systolic volume, mL	56 ± 32	56 ± 29	70 ± 39	0.069
2D ECHO end-systolic volume index, mL/m^2^	28 ± 15.0	26 ± 14	33 ± 18	0.072
2D ECHO ejection fraction, %	64 ± 6	64 ± 6	64 ± 6	0.695
3D ECHO end-diastolic volume, mL	177 ± 51	175 ± 46	196 ± 68	0.125
3D ECHO end-diastolic volume index, mL/m^2^	86 ± 23	85 ± 21	94.9 ± 28	0.108
3D ECHO end-systolic volume, mL	69 ± 24	68 ± 21	78 ± 34	0.12
3D ECHO end-systolic volume index, mL/m^2^	33 ± 11	33 ± 10	38 ± 15	0.095
3D ECHO ejection fraction, %	62 ± 5	62 ± 5	61 ± 6	0.678
MRI end-diastolic volume, mL	234 ± 81	223 ± 80	293 ± 76	˂0.001
MRI end-diastolic volume index, mL/m^2^	118 ± 30	114 ± 27	142 ± 34	˂0.001
MRI end-systolic volume, mL	88 ± 51	86 ± 41	124 ± 68	0.005
MRI end-systolic volume index, mL/m^2^	43 ± 23	41 ± 20	60 ± 28	0.003
MRI ejection fraction, %	61 ± 6	61 ± 6	60 ± 5	0.248
MRI native T1 relaxation time, ms	1023 ± 30	1023 ± 30	1022 ± 29	0.934
MRI extracellular volume fraction, %	24 ± 3	24 ± 3	24 ± 2	0.819
2D ECHO global longitudinal strain, %	−18 ± 2	−19 ± 2	−17 ± 3	0.07
2D ECHO TWIST	14 ± 4	13 ± 4	14 ± 4	0.496
3D ECHO global longitudinal strain, %	−15 ± 4	−15 ± 4	−15 ± 4	0.518
MRI global longitudinal strain, %	−15 ± 2	−15 ± 2	−14 ± 3	0.62
MRI global circumferential strain, %	−22 ± 3	−22 ± 3	−21 ± 2	0.54
MRI global radial strain, %	31 ± 7	31 ± 7	31 ± 6	0.55
*AR assessment*				
Integrative approach				0.02
Moderate-to-severe AR, N (%)	56 (54)	50 (60)	6 (30)	
Severe AR, N (%)	48 (46)	34 (40)	14 (70)	
2D ECHO vena contracta width, mm	6.5 ± 1.5	6.3 ± 1.5	6.9 ± 1.6	0.118
Diastolic flow reversal velocity, cm/s	19.4 ± 4.3	18.8 ± 4.0	22.8 ± 4.0	˂0.001
2D ECHO regurgitant volume, mL	52 ± 48	52 ± 47	69 ± 63	0.041
2D ECHO regurgitant fraction, %	36 ± 18	34 ± 18	45 ± 17	0.017
3D ECHO vena contracta area, mm^2^	29 ± 13	26 ± 11	38 ± 15	˂0.001
MRI regurgitation volume, mL	50 ± 28	44 ± 25	73 ± 30	˂0.001
MRI regurgitation fraction, %	38 ± 17	36 ± 17	49 ± 11	0.001

Values are means ± standard deviations or numbers (percentage). 2D, two-dimensional; 3D, three dimensional; AR, aortic regurgitation; ECHO, echocardiography; MRI, magnetic resonance imaging; TWIST, left ventricular torsion.

**Table 3 jcm-08-01654-t003:** Predictive accuracy of selected imaging markers to identify patients undergoing AV surgery.

	AUC (95% CI)	Cutoff Value	Sensitivity (%)	Specificity (%)
*Markers of LV remodeling*				
2D ECHO LVESD, mm	0.69 (0.57–0.80)	37 40 45	85 55 15	56 69 94
2D ECHO LVESDi, mm/m^2^	0.66 (0.54–0.78)	18 20 22	80 30 15	53 75 90
MRI LVEDV, mL	0.68 (0.53–0.83)	281	50	84
MRI LVEDVi, mL/m^2^	0.69 (0.54–0.84)	110 124 139	80 70 55	53 70 85
MRI LVESV, mL	0.64 (0.48–0.80)	121	50	84
MRI LVESVi, mL/m^2^	0.65 (0.49–0.81)	42 56 58	70 60 50	53 77 81
2D ECHO GLS, %	0.61 (0.47–0.70)	−17.5	50	71
*Markers of AR severity*				
Diastolic flow reversal velocity, cm/s	0.72 (0.59–0.85)	22	65	78
2D ECHO RV, mL	0.58 (0.43–0.74)	93	40	88
2D ECHO RF, %	0.61 (0.47–0.76)	47	50	73
3D VCA (mm^2^)	0.76 (0.64–0.88)	29 31 36	80 75 65	63 71 81
MRI RV, mL	0.79 (0.70–0.88)	41 45	95 90	56 61
MRI RF, %	0.77 (0.68–0.86)	34	95	55
*Integrative approach*				
2D ECHO integrative approach	0.65 (0.52–0.78)	Severe AR	70	60
3D ECHO VCA ≥30 mm^2^ and 2D ECHO LVESD or LVESDi	NA	LVESD >40 mm LVESD >45 mm LVESDi >20 mm/m^2^ LVESDi >22 mm/m^2^	80 80 80 80	71 97 77 87
MRI regurgitant volume ≥45 mL and MRI LVEDVi or LVESVi	NA	LVEDVi >139 mL/m^2^ LVESVI >62 mL/m^2^	90 90	78 78
MRI regurgitant fraction ≥34% and MRI LVEDVi or LVESVi	NA	LVEDVi >139 mL/m^2^ LVESVI >62 mL/m^2^	95 95	89 89

2D, two-dimensional; 3D, three dimensional, AUC, area under curve; ECHO, echocardiography; GLS, global longitudinal strain; LV, left ventricle; LVEDV, left ventricular end-diastolic volume; LVEDVi, left ventricular end-diastolic volume index; LVESD, left ventricular end-systolic diameter; LVESDi, left ventricular end-systolic diameter index; LVESV, left ventricular end-systolic volume; LVESVi, left ventricular end-systolic volume index; MRI, magnetic resonance imaging; RF, regurgitant fraction; RV, regurgitant volume; VCA, vena contracta area.

**Table 4 jcm-08-01654-t004:** Independent predictors of aortic valve surgery.

	Univariable Analysis		Multivariable Analysis	
	HR (95% CI)	*p*-Value	HR (95% CI)	*p*-Value
2D ECHO LVEDD	1.08 (0.99−1.18)	0.084		
2D ECHO LVESD	1.12 (1.02−1.23)	0.014	1.12 (1.02−1.23)	0.018 *
2D ECHO LVESDi	1.18 (1.02−1.37)	0.031	1.18 (1.01−1.38)	0.042 *
MRI LVEDV	1.01 (1.00−1.02)	0.004	1.01 (1.00−1.01)	0.036 †
MRI LVEDVi	1.02 (1.00−1.03)	0.004	1.01 (1.00−1.03)	0.033 †
MRI LVESV	1.02 (1.00−1.03)	0.017		
MRI LVESVi	1.03 (1.01−1.06)	0.014		
2D ECHO RV	1.01 (1.00−1.02)	0.011	1.01 (1.00−1.02)	0.035 ‡
2D ECHO RF	1.03 (1.01−1.06)	0.018	1.03 (1.00−1.06)	0.020 ‡
3D VCA	1.07 (1.04−1.10)	<0.001	1.06 (1.03−1.10)	<0.001 ‡
MRI RV	1.03 (1.02−1.05)	<0.001	1.03 (1.01−1.04)	<0.001 §
MRI RF	1.05 (1.03−1.08)	<0.001	1.05 (1.02−1.08)	<0.001 §

CI, confidence interval; HR, hazard ratio; for other abbreviations see previous tables. * LVESD and LVESDi remained significant predictors of aortic valve (AV) surgery after adjustment for ECHO RV and ECHO RF but they lost predictive significance in combination with 3D VCA. † MRI LVEDV and LVEDVi showed borderline significance to predict AV surgery after adjustment with MRI RF but they lost predictive significance in combination with MRI RV. ‡ 2D ECHO RV, 2D ECHO RF and 3D VCA consistently retained their independent predictive value after adjustment for ECHO-derived LV diameters or their indices, 3D VCA was the strongest predictor. § MRI RV and RF were strong independent predictors after adjustment for MRI-derived LV volumes or their indices.

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
