# Peer review of "New Imaging Markers of Clinical Outcome in Asymptomatic Patients with Severe Aortic Regurgitation"

_jcm, 2019, doi:10.3390/jcm8101654_

Round 1

Reviewer 1 Report

This is an interesting article to address potential biomarker of AR. There are several issues need to be taken care:

Imaging parameters such as Parameters related to LV morphology and function,  3D ECHO-derived VCA38 and MRI-derived RV and RF are not biomarkers, the title and content in the manitext has to be modified. A classical biomarker is some molecules such as IL-6, hs-CRP, IL1b etc means those in circulating blood or body fluid.  Introduction: too short, introduce the pathology mechanism of AR briefly Description: "This is a table with baseline clinical characteristic" is not scientific. Other tables have the same issue Advantages over traditional imaging techniques can be made stronger.

Author Response

Point 1: Imaging parameters such as Parameters related to LV morphology and function,  3D ECHO-derived VCA38 and MRI-derived RV and RF are not biomarkers, the title and content in the manitext has to be modified. A classical biomarker is some molecules such as IL-6, hs-CRP, IL1b etc means those in circulating blood or body fluid.  

 Response 1: Thank you for this comment. We have changed the title –

New Imaging Markers of Clinical Outcome in Asymptomatic Patients with Severe Aortic Regurgitation

Point 2: Introduction: too short, introduce the pathology mechanism of AR briefly

Response 2: We acknowledge the validity of this comment. We have entered a brief information about the AR pathology.

Chronic aortic regurgitation (AR) is the third most common valvular heart disease in Western countries, with prevalence of 0.1% to 2.0% Degenerative aetiology on tricuspid or bicuspid aortic valve and annuloaortic ectasia are the most common causes of chronic AR.  Rheumatic fever remains a frequent cause of chronic AR in developing countries. [1-4]. In chronic AR, progressive left ventricular (LV) dilatation compensates for the increase of LV end-diastolic pressure. Left ventricular dilatation enables to preserve cardiac output despite the large regurgitant volume of blood returning to the LV in diastole. Increased afterload in chronic AR is compensated by eccentric LV hypertrophy. This complex volume and pressure compensatory mechanism explain the long asymptomatic course of the disease.

Point 3: Description: "This is a table with baseline clinical characteristic" is not scientific. Other tables have the same issue Advantages over traditional imaging techniques can be made stronger.

Response 3: Thank you for this useful comment. The text has been changed accordingly.

“Baseline clinical characteristics”

“Baseline imaging characteristics”

“Predictive accuracy of selected imaging markers to identify patients undergoing AV surgery”

“Independent predictors of aortic valve surgery”

Reviewer 2 Report

In this manuscript authors have demonstrated that new imaging biomarkers as early stage diagnosis in patients with severe aortic valve regurgitation. 

Also, authors have demonstrated that 3D ECHO-derived VCA and MRI-derived RV and RF showed high accuracy and excellent sensitivity to identify patients in need for early AV surgery over LV morphology and function which has showed moderate accuracy.

Most often, aortic valve regurgitation develops gradually, may have no signs or symptoms for years, and may even be unaware that the patient has the condition. Despite, many studies have already been demonstrated the use of 3D Echo evaluation in various conditions; Mitral Valve Regurgitant, Pulmonary Hypertension, etc., this study demonstrated the use of 3D ECHO evaluation in aortic valve regurgitation patients.

The manuscript has well written, and the study design and criteria for patients selection is acceptable.

This study is useful in clinical settings. In Figure 1, C, D and F are not clear. I can’t see any change in the circle and the outside.

Author Response

Point 1: This study is useful in clinical settings. In Figure 1, C, D and F are not clear. I can’t see any change in the circle and the outside.

Response 1: Thank you very much for your comment, we acknowledge its validity. We hope you find the re-written Figure 1 legend satisfactory.

Figure 1.  New imaging markers. A) Echocardiography derived three-dimensional vena contracta area, B) Echocardiography two-dimensional global longitudinal strain, C) Magnetic resonance – the left ventricular outflow tract (cine), red line – through-plane flow sequence slice position displayed on D, Ao – aorta, LA – left atrium, LV – left ventricle, RV – right ventricle, D) Through-plane flow sequence at sinotubular junction level (STJ) of the aorta (displayed on C), the blue circle is a manually drawn region of interest where the blood flow and regurgitant volume and fraction are calculated, the exact copy of the region interest is in all four images, Phantom – stationary phantom useful for flow measurement correction, E) Flow-time curve based on (D) – blue line shows blood flow at STJ and red line show flow in stationary phantom, F) Native T1 mapping from MOLLI sequence, blue circle – a semi-automatically drawn region of interest within the blood pool, blue ellipsoid – a manually drawn region of interest within the myocardium at the level of the interventricular septum utilized for myocardial fibrosis calculation.

Reviewer 3 Report

The study is well-done and the manuscript is well-written.

The study seeks to identify more sensitive imaging biomarkers of AV intervention for asymptomatic AR patients, and discovered 3D ECHO-derived VCA and MRI-derived RV and RF to be most sensitive with decent specificity.

Below are my comments.

What is the difference between “accuracy” and “specificity” in the current manuscript? The newly identified biomarkers showed good sensitivity but are less specific than 2D-derived parameters. Please comment on how these newly identified biomarkers can be applied to the clinic. In other words, under what circumstances are the newly identified biomarkers more suitable to predict AV intervention than conventional biomarkers and vice versa.

Author Response

Point 1: What is the difference between “accuracy” and “specificity” in the current manuscript? The newly identified biomarkers showed good sensitivity but are less specific than 2D-derived parameters.

Response 1: Thank you very much for your question.

Specificity in our manuscript means the ability of a particular biomarker to identify all patients with true fast disease progression

sensitivity

Accuracy in our manuscript means the ability of a particular biomarker to identify (or distinguish) patients with fast or slow disease progression. Some markers are highly sensitive but also has only mild or modest specificity. These markers are not reliable or accurate.

Point 2: Please comment on how these newly identified biomarkers can be applied to the clinic. In other words, under what circumstances are the newly identified biomarkers more suitable to predict AV intervention than conventional biomarkers and vice versa.

Response 2: Thank you very much for this comment. The clinical application needs to be clarified.

Our new imaging biomarkers such as 3D ECHO-derived VCA and MRI-derived RV and RF can be used in all patients with asymptomatic severe chronic aortic regurgitation to identify patients at the risk of early disease progression requiring early surgical intervention.

Round 2

Reviewer 1 Report

The authors well addressed previous concerns, and is now acceptable